# Risk Factors for Non-Union after Open Reduction and Internal Fixation in Patients with Distal Humerus Fractures

**DOI:** 10.3390/jcm11102679

**Published:** 2022-05-10

**Authors:** Ki-Hyeok Ku, Jong-Hun Baek, Myung-Seo Kim

**Affiliations:** 1Shoulder & Elbow Clinic, Department of Orthopaedic Surgery, College of Medicine, Kyung Hee University Hospital at Gangdong, Seoul 05278, Korea; bestdoctorku@gmail.com; 2Shoulder & Elbow Clinic, Department of Orthopaedic Surgery, College of Medicine, Kyung Hee University Hospital, Seoul 02447, Korea; paeton81@naver.com

**Keywords:** distal humerus fracture, the open reduction and internal fixation, non-union, risk factors

## Abstract

Background: Only a few studies have reported on the risk factors for non-union after open reduction and internal fixation (ORIF) in distal humerus fractures. Methods: We retrospectively reviewed 155 patients who underwent ORIF for distal humerus fractures from January 2008 to June 2020. Various patient factors, including body mass index (BMI), diabetes mellitus (DM), and combined fracture, as well as surgical factors, including fixation methods (e.g., orthogonal plate/parallel plate/single plate/tension bend wiring [TBW]) and combined fracture operations, were evaluated as risk factors for non-unions. Results: Among the patient factors, BMI (25.0 ± 3.4 vs. 22.7 ± 3.4, *p* = 0.032), DM (5/13 [38.5%] vs. 20/142 [14.1%], *p* = 0.038) and combined fracture (5/13 [38.5%] vs. 16/142 [11.3%], *p* = 0.018) were significantly different between groups with non-union and union. Among the surgical factors, combined fracture operation (5 [38.5%] vs. 9 [6.3%], *p* = 0.002) and the fixation method (3 [23.1%]/1 [7.7%]/4 [30.8%]/5 [38.5%] vs. 84 [59.2%]/7 [4.9%]/40 [28.2%]/11 [7.7%], *p* = 0.005) showed a significant difference between groups with non-union and union. Multivariate regression analysis showed that combined fracture operation (OR 10.467; 95% CI 1.880–58.257; *p* = 0.007) and TBW (OR 9.176; 95% CI 1.474–57.135; *p* = 0.018) among the fixation methods posed as a significant risk factor for non-union. Conclusions: The risk of non-union increased in patients who underwent surgery for another fracture combined with distal humerus fracture and in patients who underwent ORIF with TBW.

## 1. Introduction

Although distal humerus fractures account for a small percentage of all fractures [1], high complication rates have been reported after surgical treatment due to fracture complexity and poor bone quality [2,3]. It is reported that up to 25% of non-unions occur after distal humerus fracture surgery [4,5]; in most cases, further operative treatment with revision ORIF or total elbow arthroplasty is necessary [5,6]. Vauclair et al. reported that when additional surgery is performed for non-union, a very high complication rate and unsatisfactory results may still occur [5]. In their study, total elbow arthroplasty was performed in 3 of 15 patients with non-union of distal humerus fracture, as fracture healing did not occur despite revision ORIF and autogenous bone grafting. They noted that the identification of risk factors for non-unions before surgery was important to improve the fracture healing process but did not analyze it.

ORIF is the treatment of choice for distal humerus fractures [7]. However, Ring et al. reported that the unstable non-union rate can increase by up to 75% if each column cannot be rigidly fixed using plates [8]. Claessen et al. analyzed factors related to reoperation after ORIF in distal humerus fractures and reported that a high Charlson Comorbidity Index, obesity, diabetes mellitus (DM), osteoarthritis, and smoking were risk factors for reoperation [9]. However, in this study, nonunion risk factors were not analyzed alone and the fixation method, which has an important effect on stability in ORIF of distal humerus fractures, was not mentioned in detail. This study aimed to evaluate the independent risk factors for non-union, including various patient and surgical factors (e.g., fixation method), after ORIF in patients with distal humerus fracture.

## 2. Materials and Methods

This retrospective case-control study investigated the risk factors for non-union after ORIF in patients with distal humerus fractures. This study was approved by the institutional review board and informed consent was obtained from all patients.

### 2.1. Patient Selection and Methods

We retrospectively reviewed patients who underwent ORIF for distal humerus fractures at two hospitals (Kyung Hee University Hospital, Seoul, Korea and Kyung Hee University Hospital at Gangdong, Seoul, Korea) between January 2008 and June 2020. All surgeries were performed by two attending elbow trauma surgeons (P.J.H. and L.J.H.) in each hospital. We included patients over 18 years old with skeletal maturity [10], who underwent ORIF for AO classification type A, B, and C distal humerus fractures and were followed up for at least 1 year. The exclusion criteria were as follows: pathologic fracture, open fracture of Gustilo and Anderson classification ≥2 [11], patients with hemiplegia in the ipsilateral upper extremity, and loss to follow-up.

During the study period, 172 patients received operative treatment for distal humerus fractures, and 155 patients were enrolled after the inclusion and exclusion criteria were applied. Of these, 3 patients had pathologic fractures, 2 patients had open fractures (Gustilo and Anderson classification ≥2), 1 patient had hemiplegia of the ipsilateral upper extremity, and 11 patients lost to follow-up met our exclusion criteria.

Radiologic union was defined as bridging trabeculae observed in at least three of the four cortices at the fracture site on biplanar plain radiographs [6]. Non-union was defined as the absence of any bridging trabeculae between the fracture ends on a plain radiograph six months after surgery [12].

### 2.2. Patient Demographic Factors

Demographic data were collected from inpatient medical records before surgery: age, sex, side (right or left), body mass index (BMI), various comorbidities (e.g., hypertension, DM, angina or myocardial infarction, and cancer), smoking history, alcohol history, injury mechanism, time between injury and operative treatment, combined fractures in a different location at the time of injury, Gustilo and Anderson classification 1, and preoperative neurological symptoms. According to Robinson et al., a simple fall from a standing height or below was classified as low-energy and a fall from a greater height, a road traffic accident, sports, or others was classified as a high-energy mechanism [13].

### 2.3. Surgical Factors

Using three-dimensional computed tomography performed within one week of surgery, the AO classification type of distal humerus fracture was classified into A, B, and C, and the following surgery-related data were collected through medical records: combined fracture operation, surgical approach, fixation method, ulnar nerve anterior transposition, and operation time. Surgical approaches were classified as (1) posterior approach, (2) medial approach, (3) lateral approach, and (4) combined medial-lateral approach. The posterior approach was subdivided into (1) olecranon osteotomy, (2) triceps split, and (3) triceps-sparing paratricipital approach. Fixation methods were classified into the orthogonal plate, parallel plate, single plate, and tension band wiring (TBW).

### 2.4. Operative Techniques and Rehabilitation

All surgeries were performed in the supine position under general anesthesia using a tourniquet. Each approach was performed using standard methods [14,15,16,17]. In the posterior approach, the ulnar nerve was first identified and protected. In cases of severe intra-articular fracture fragmentation, olecranon chevron osteotomy is performed. Subsequently, depending on the surgeon’s preference, either orthogonal, parallel, or single 3.5 LCP distal humerus plate (DuPey Synthes, West Chester, PA, USA) fixation or TBW fixation was performed (Figure 1A–D). Anatomical reduction was implemented as much as possible to achieve absolute stability, and at least three cortical or locking screws were inserted proximally and distally.

In the case of the combined mediolateral approach, as in Wei et al. [18], the medial incision was performed first and the ulnar nerve was identified but not released. A reduction clamp was used to hold the medial side of the proximal and distal fragments, and one or two Kirschner wires were used for temporary fixation. Subsequently, the lateral side of the fracture fragment was approached through a lateral incision and temporary fixation was performed in the same way, and then a 3.5 LCP distal humerus plate (DuPey Synthes, West Chester, PA, USA) was applied either orthogonally or parallelly (Figure 2A–D).

All patients were administered a long-arm splint with the elbow joint at 90° flexion for up to 2 weeks after surgery and then started active-assisted motion exercises.

### 2.5. Statistical Analysis

The Mann–Whitney U test, Wilcoxon signed-rank test, and chi-squared or Fisher’s exact test were performed to compare the patient demographic and surgical factors. Any significant predictive values in the univariate analysis were included in the multivariate logistic regression analysis to identify independent risk factors for non-union. Significance was set at *p* = 0.05, with associated 95% confidence intervals (CIs). The SPSS software package (version 21.0; IBM Corp, Armonk, NY, USA) was used for all statistical analyses. Post hoc power calculation (alpha error: 0.05, 1-beta error: 0.95) and priori analysis were performed using G * Power (version 3.1.9.7; Heinrich-Heine-Universität Düsseldorf, Düsseldorf, Germany).

## 3. Results

The demographic data and surgical factors of the 155 enrolled patients are summarized in Table 1. The duration required for union in the union group was 22.9 ± 15.8 weeks, and non-union was observed in 13 patients (8.4%).

### 3.1. Comparison of Demographic Factors between Non-Union Group and Union Group

The BMI was significantly higher in the non-union group than in the union group (25.0 ± 3.4 vs. 22.7 ± 3.4, *p* = 0.032). The proportion of patients with DM was also significantly higher in the non-union group than in the union group (5/13 [38.5%] vs. 20/142 [14.1%], *p* = 0.038). In addition, combined fractures were more common in non-union patients than in union patients (5/13 [38.5%] vs. 16/142 [11.3%], *p* = 0.018). Other patient demographic factors showed no significant differences between the two groups (Table 2).

### 3.2. Comparison of Surgical Factors between Non-Union Group and Union Group

Compared with the union group, the frequency of combined fracture operations was significantly higher in the non-union group (5 [38.5%] vs. 9 [6.3%], *p* = 0.002) (Table 3). Regarding the fixation method, in the non-union group, the orthogonal plate was used in 3 patients (23.1%), the parallel plate in 1 patient (7.7%), the single plate in 4 patients (30.7%), and TBW in 5 patients (38.5%). In the union group, the orthogonal plate was performed in 84 patients (59.2%), parallel plates in 7 patients (4.9%), single plates in 40 patients (28.2%), and TBW in 11 patients (7.7%), which showed a significant difference between the two groups (*p* = 0.005) (Table 4). A post hoc power analysis was conducted using the fixation method, which showed a statistically significant difference and was the most important value. The effect size was 0.9063357 and the alpha error was 0.05; the power of this study was 0.99. The required sample size was 14 in the total group, indicating that the sample size was sufficient to have a power >0.80.

### 3.3. Independent Risk Factors for Non-Union

A multivariate logistic regression analysis showed that combined fracture operation (odds ratio 10.467; 95% CI 1.880–58.257; *p* = 0.007) and TBW (odds ratio 9.176; 95% CI 1.474–57.135; *p* = 0.018) were independent risk factors for non-union after ORIF in distal humerus fracture (Figure 3 and Table 5).

Of the 13 patients with non-union, revision ORIF with auto bone graft was performed in eight patients and fracture union was obtained. One patient underwent total elbow arthroplasty, 1 patient had functional non-union, 1 patient refused additional surgery, and 2 patients were lost to follow up while following up at an outpatient clinic after diagnosis of non-union.

## 4. Discussion

In this study, non-union occurred in 8.4% (13/155) of distal humerus fractures after ORIF. The combined fracture operation and fixation method showed that TBW was an independent risk factor for non-union.

The gold standard treatment for distal humerus fractures is ORIF [7], but internal fixation is challenging because of the complex anatomy of the elbow and relative osteopenia [19]. Particularly in elderly patients, stable internal fixation is not guaranteed owing to osteoporotic bones, metaphyseal comminution, and poor soft-tissue conditions [20]. For these reasons, the non-union rate in distal humerus fractures remains high despite advances in surgical techniques [4,5].

Vanclair et al. reported that after ORIF of humerus fractures, non-unions occurred more frequently in the distal regions than in the proximal and diaphyseal regions, and the incidence of non-union after distal humerus fracture has been reported as up to 25% [5]. Moursy et al. reported that non-union occurred in 22% of patients with distal humerus fractures of AO classification A, B, and C after ORIF in elderly patients over 70 years of age [2]. In this study, non-union occurred in approximately 8% of the patients and the rate was lower than that in previous studies, which is thought to be because the average age of patients enrolled in this study was relatively young.

Previously reported risk factors for fracture non-union in patient demographics included advanced age, smoking, obesity, and metabolic diseases such as DM [21,22]. In addition, Zura et al. reported that in their study analyzing various patient factors related to overall fracture non-union, multiple fractures were one of the risk factors for non-union [22]. However, only a few studies have reported on the non-union risk factors for distal humerus fractures and the overall fracture non-union risk factors. Ali et al. reported that approximately 40% of patients who had non-union among those who underwent ORIF for distal humerus fracture were current smokers [23]. However, the number of patients enrolled in the study was small and the risk factors were not analyzed. Claessen et al. analyzed the risk factors for reoperation after ORIF of distal humerus fractures due to early loosening or breakage of implants or non-union and reported that greater comorbidities, obesity, DM, and smoking were associated with reoperation [9]. However, the independent risk factors for non-union were not evaluated in this study. In this study, univariate analysis showed differences in DM, BMI, and combined fracture in the non-union and union groups, but in multivariate logistic regression analysis, they were not independent risk factors for non-union.

Since the anatomical LCP system plate has been used, more stable fixation of distal humerus fractures can be obtained [19], and Ali et al. suggested that the initial fixation method should be considered as a surgical factor related to non-union of distal humerus fractures [23]. In the above study, inadequate ORIF, such as using only Kirschner wires or screws, was performed as the primary operation in 75% of the 16 patients with fracture non-union. In addition, another study reported that the unstable non-union rate could be very high if each column is not firmly fixed [8]. However, few studies have analyzed various fixation methods for distal humerus fractures. In the current study, we analyzed surgical factors, including various fixation methods performed during ORIF for distal humerus fractures. The results showed that combined fracture operation and TBW were independent risk factors for non-union and that when TBW was performed, the risk of non-union increased by about nine times compared to fixation with orthogonal plates.

In several previous studies, a good union rate was reported after fixation of a distal humerus fracture with TBW [24,25]. Rahman et al. reported that unions were obtained in all patients after TBW for intercondylar humerus fractures [25]. Kokly et al. also reported that fracture site unions were obtained in all patients with distal humerus fractures after TBW [24]. However, in both studies, the number of enrolled patients was small, only analyzing 25 patients. In addition to the above studies, no clinical studies have compared TBW and dual-plate fixation in distal humerus fractures. Although not a clinical study, Doğramaci et al. reported that dual-plate fixation had superior stability to TBW in their biomechanical study and reported similar results to this study [26].

This study has several limitations. First, although it was a study that analyzed the risk factors for fracture non-unions, other complications excluding non-unions were not analyzed. Second, although the AO classification type of distal humerus fracture was not a non-union risk factor, types A, B, and C were analyzed as one group. Third, conservative treatment can be performed in selected patients with distal humerus fractures. Since this study was conducted on patients who underwent ORIF for distal humerus fracture, we could not evaluate the risk factors for non-union that could occur after conservative treatment. Despite these limitations, this study is valuable in that prior studies did not assess the independent risk factors for non-union after ORIF in distal humerus fractures.

## 5. Conclusions

Non-union occurred in approximately 8% of the patients who underwent ORIF for distal humerus fractures. The risk of non-union increased in patients who underwent combined fracture operations and ORIF with TBW fixation.

## Figures and Tables

**Figure 1 jcm-11-02679-f001:**
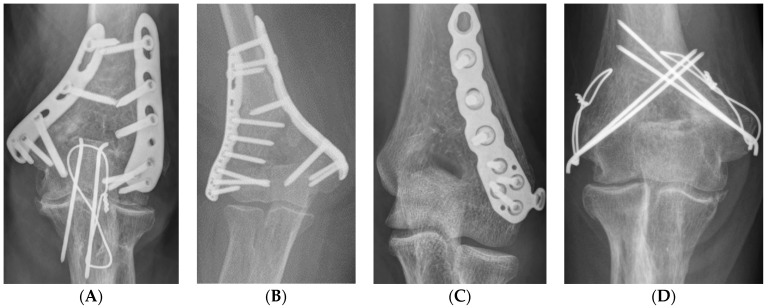
Fixation method (**A**) orthogonal plate, (**B**) parallel plate, (**C**) single plate, (**D**) tension band wiring.

**Figure 2 jcm-11-02679-f002:**
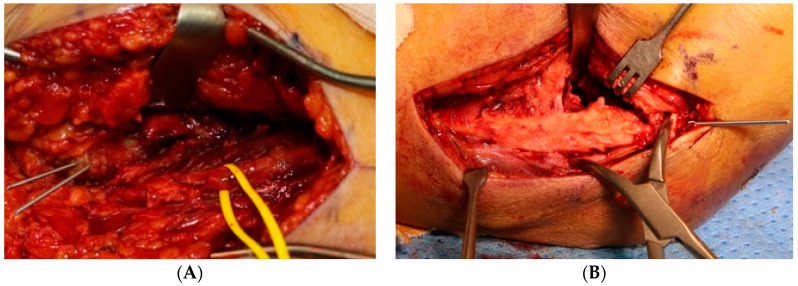
Dual incision (**A**) medial approach, identification of ulnar nerve (arrow), temporary fixation using two Kirschner wires, (**B**) lateral incision, temporary fixation using one Kirschner wire, (**C**) intraoperative C-arm radiograph, (**D**) orthogonal plate fixation.

**Figure 3 jcm-11-02679-f003:**
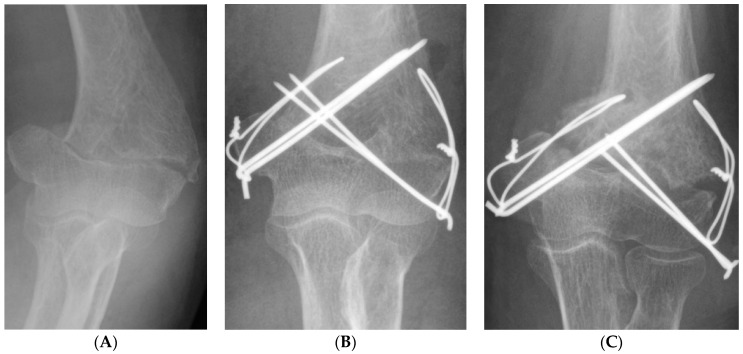
A 67-year-old female patient with non-union after ORIF due to distal humerus fracture with usage of tension band wiring as the fixation method (**A**) pre-operative simple radiograph, (**B**) immediate post-operative simple radiograph, (**C**) a non-union was observed in the simple radiograph 7 months after the surgery.

**Table 1 jcm-11-02679-t001:** Initial patient demographic and surgical factors.

Demographic Factors	
No. of patients	155
Age, years	52.6 ± 21.3
Sex, male/female, *n*	53/102
Side, right/left, *n*	62/93
BMI (kg/m^2^)	22.9 ± 3.4
Comorbidities, *n*	
Hypertension	50
Diabetes mellitus, HbA1c level	25, 7.6 ± 0.9
Angina or myocardial infarction	8
Cancer	4
Smoking, *n*	16
Alcohol, *n*	34
Injury mechanism, low/high	95/59
Injury to operation interval (days)	6.1 ± 6.6
Combined fracture, *n*	21
Gustilo and Anderson classification 1	5
Preop neurologic symptom	9
Follow up period, months (range)	33.9 ± 30.1 (22–312)
Non-union, *n*	12
Surgical factors	
AO fracture classification, A/B/C	68/11/76
Combined fracture operation, *n*	14
Surgical approach, posterior/medial/lateral/dual incision, *n*	118/4/27/6
Olecranon osteotomy/triceps split/triceps sparing paracipital ‡, *n*	87/14/17
Fixation method, orthogonal plate/parallel/single/TBW, *n*	87/8/44/16
Ulnar nerve anterior transposition, *n*	76
Operation, time, minutes	124.2 ± 50.5

BMI, body mass index; Preop, preoperative; AO, Arbeitsgemeinschaft für Osteosynthesefragen; TBW, tension band wiring. ‡ Among the posterior incisions.

**Table 2 jcm-11-02679-t002:** Comparison of demographic factors between non-union patients and union patients.

Demographic Factors	Non-Union (*n* = 13)	Union (*n* = 142)	*p* Value
Age, years	63.1 ± 19.4	51.7 ± 21.3	0.065
Sex, male/female, *n*	7/6	46/96	0.135
Side, right/left, *n*	6/7	56/86	0.636
BMI (kg/m^2^)	24.9 ± 3.2	22.7 ± 3.4	0.028
Comorbidities, *n* (%)			
Hypertension	6	44	0.352
Diabetes mellitus	5 (38.5)	20 (14.1)	0.038
Angina or myocardial infarction	1	7	0.512
Cancer	0	4	1.000
Smoking, *n* (%)	2 (15.4)	14 (9.9)	0.626
Alcohol, *n* (%)	2 (15.4)	32 (22.5)	0.735
Injury mechanism, low/high, *n*	8/5	87/54	1.000
Injury to operation interval (days)	7.3 ± 11.8	6.0 ± 6.0	0.728
Combined fracture, *n*	5	16	0.018
Gustilo and Anderson classification 1	1	4	0.359
Preop neurologic symptom, *n*	1	8	0.555
Follow-up period, months	37.2 ± 27.1	33.5 ± 30.9	0.678

BMI, body mass index; Preop, preoperative.

**Table 3 jcm-11-02679-t003:** Comparison of surgical factors between non-union patients and union patients.

Surgical Factors	Non-Union (*n* = 13)	Union (*n* = 142)	*p* Value
AO fracture classification, A/B/C, *n*	8/1/4	60/10/72	0.371
Combined fracture operation, *n* (%)	5 (38.5)	9 (6.3)	0.002
Ulnar nerve anterior transposition, *n* (%)	3 (23.1)	73 (51.4)	0.080
Operation, time, minutes	93.8 ± 32.2	125.9 ± 50.9	0.131

**Table 4 jcm-11-02679-t004:** Comparison of surgical approaches and fixation methods between non-union patients and union patients.

Surgical Factors	Non-Union (*n* = 13)	Union (*n* = 142)	*p* Value
Surgical approach, *n*			
Posterior	9	109	0.170
Medial	0	4	
Lateral	2	25	
Dual incision	2	4	
Olecranon osteotomy/triceps split/paracipital ‡	8/1/0	79/13/17	0.626
Fixation method, *n* (%)			
Orthogonal plate	3 (23.1)	84 (59.2)	0.005
parallel	1 (7.7)	7 (4.9)	
single	4 (30.8)	40 (28.2)	
TBW	5 (38.5)	11 (7.7)	

TBW, tension band wiring. ‡ Among the posterior incisions.

**Table 5 jcm-11-02679-t005:** Independent risk factors for non-union after ORIF in patients with distal humerus fracture.

Multivariate Analysis	Odds Ratio	95% CI	*p* Value
BMI	1.092	0.893–1.335	0.393
Diabetes mellitus	3.528	0.864–14.403	0.079
Combined fracture	2.464	0.266–22.855	0.427
Combined fracture operation	10.467	1.880–58.257	0.007
Fixation method §			
Parallel plate (ref. orthogonal)	5.850	0.422–81.107	0.188
Single plate (ref. orthogonal)	5.200	0.889–30.430	0.067
TBW (ref. orthogonal)	9.176	1.474–57.135	0.018

ORIF, open reduction and internal fixation; CI, confidence interval; BMI, body mass index; TBW, tension band wiring. § orthogonal plate, parallel plate, single plate, and TBW as revealed by the generalized linear model.

## Data Availability

The authors agree to share their raw data, any digital study materials, and analysis codes as appropriate.

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
