# Peer review of "Risk Factors for Non-Union after Open Reduction and Internal Fixation in Patients with Distal Humerus Fractures"

_jcm, 2022, doi:10.3390/jcm11102679_

Round 1

Reviewer 1 Report

The relevance of the subject treated here is great in view of the aging structures in the population. In this respect, it can be assumed that reading material was presented that aroused sufficient interest for many readers. Even if the study design is retrospective, the conception of the study is suitable for answering the original questions. The size of the collective is representative, the statistical elaboration is reliable and well written. The authors have observed the techniques that are currently used in the therapy of distal humeral fractures over a long period of time and included relevant risks as influencing factors. Exclusion factors are categorized and the conclusions are presented in an easy to understand way. Overall, the study can therefore be used in everyday clinical practice as an aid for risk stratification of failure. Ultimately, only the bone quality is missing in the assessment of the risks, which, however, is dispensable given the retrospective design and the young patient collective.

Author Response

The relevance of the subject treated here is great in view of the aging structures in the population. In this respect, it can be assumed that reading material was presented that aroused sufficient interest for many readers. Even if the study design is retrospective, the conception of the study is suitable for answering the original questions. The size of the collective is representative, the statistical elaboration is reliable and well written. The authors have observed the techniques that are currently used in the therapy of distal humeral fractures over a long period of time and included relevant risks as influencing factors. Exclusion factors are categorized and the conclusions are presented in an easy to understand way. Overall, the study can therefore be used in everyday clinical practice as an aid for risk stratification of failure. Ultimately, only the bone quality is missing in the assessment of the risks, which, however, is dispensable given the retrospective design and the young patient collective.

Response: Thank you for your thoughtful comment.

Reviewer 2 Report

Thank you for allowing me to review your work. Overall, i think this article is of merit as it includes a fairly large number of distal humerus fractures and analyzes risk factors of non union.

Introduction:

Please give more details on the cited studies (number of cases, follow up etc) as well as more details on some claims made (how many patients end up with a total elbow arthroplasty?).

Furthermore, as i think surgical factors are very important as we can influence them relevantly, please give more details on surgical approaches (single, double plating, additional third plate, plate positioning etc. , surgical appoarch etc) and the associated adavantages/disadvantages when it comes to non-union rates.

Methods: what percentage of patient undwent non-operative treatement or proceeded directly to artrhoplasty?

Did you use external fixation in any case as an interim or definitive surgery?

Was there a minimum follow up and what was your algorith  to follow-up? Did you do routine CT scans postoperatively?

What do you mean by combined fracture? More than one fracture at the time of surgery in a different location?  Try to reword.

Was there any association with first degree open fractures and outcome? 

The table on surgical facturos is hard to follow. Please provide separate tables on the approacehs and fixation.

How did you manage non-union in this cohort`? Please elaborate.

Where there any infections or infected non.-unions? How did you diagnose and treat those?

Discussion: Please provide more details on the studies you cite and where your study comes in. I still do not know what the studies by Vanclair or Claessen are about.

Please restructure the discussion.

One paragraph non-union rates in different studies

one paragraph patient risk factors

one paragraph surgical risk factors

one paragraph limitations. As discussed above there are quite a few more.

Author Response

Thank you for allowing me to review your work. Overall, i think this article is of merit as it includes a fairly large number of distal humerus fractures and analyzes risk factors of non union.

Introduction:

Please give more details on the cited studies (number of cases, follow up etc) as well as more details on some claims made (how many patients end up with a total elbow arthroplasty?).

Response: Thank you for your comment. We added more details of the cited studies as the reviewer’s opinion.

Page 1, lines 36-38: In their study, total elbow arthroplasty was performed in three of 15 patients with nonunion of distal humerus fracture, as fracture healing did not occur despite revision ORIF and autogenous bone grafting.

Furthermore, as i think surgical factors are very important as we can influence them relevantly, please give more details on surgical approaches (single, double plating, additional third plate, plate positioning etc. , surgical appoarch etc) and the associated adavantages/disadvantages when it comes to non-union rates.

Response: Thank you for your comment. We agreed with the reviewer’s opinion. Unfortunately, however, there are only few studies reporting the relationship between the surgical approach or fixation method and nonunion of distal humerus fracture. We have already described the study about the relationship between surgical approach and nonunion of distal humerus fracture in more detail in the discussion part.

Page 8, lines 222-226: Ali et al. suggested that the initial fixation method should be considered as a surgical factor related to nonunion of distal humerus fractures [23]. In the above study, inadequate ORIF, such as using only Kirschner wires or screws, was performed as the primary operation in 75% of the 16 patients with fracture nonunion.

Methods: what percentage of patient undwent non-operative treatement or proceeded directly to artrhoplasty?

Response: Thank you for your comment. We retrospectively reviewed patients who underwent ORIF for distal humerus fracture in this study. Therefore, the percentage of patients who underwent non-operative treatment was unknown, and we added this as a limitation of this study..Total elbow arthroplasty was performed directly for distal humerus fracture in 6 patients, accounting for about 3.4% of all patients. This study analyzed the risk factors related to the nonunion of distal humerus fracture and the percentage was not mentioned in the manuscript.

Did you use external fixation in any case as an interim or definitive surgery?

Response: Thank you for your comment. We retrospectively reviewed the patient’s operation record again, and none of the patients underwent external fixation through interim or definitive surgery.

Was there a minimum follow up and what was your algorith  to follow-up? Did you do routine CT scans postoperatively?

Response: Thank you for your comment. We added follow up range including minimum follow up period to page 5, table 1, according to the reviewer’s comment. We did not perform postoperative CT as a routine.

Page 5, table 1:

Demographic factors

Follow up period, months (range)

33.9±30.1 (22–312)

What do you mean by combined fracture? More than one fracture at the time of surgery in a different location?  Try to reword

Response: Thank you for your comment. The reviewer’s opinion was correct and we reworded the sentence for combined fracture according to reviewer’s opinion.

Page 2, line 79-80: combined fractures in a different location at the time of injury

Was there any association with first degree open fractures and outcome? 

Response: Thank you for your comment. We apologize for the omission of data on first degree open fractures. We added data about the relationship between Gustilo and Anderson classification 1 and nonunion to the method part, Table 1 and 2.

Page 2, line 80: Gustilo and Anderson classification 1

Page 5, Table 1:

Demographic factors

Gustilo and Anderson classification 1

5

Page 6, Table 2:

Demographic factors

Non-union

(n=13)

Union

(n=142)

P value

Gustilo and Anderson classification 1

1

4

0.359

The table on surgical facturos is hard to follow. Please provide separate tables on the approacehs and fixation.

Response: Thank you for your comment. We added Table 4 by separating approaches and fixation method according to the reviewer’s opinion, and changed the existing Table 4 to 5.

Table 3. Comparison of surgical factors between non-union patients and union patients.

Surgical factors

Non-union

(n=13)

Union

(n=142)

P value

AO fracture classification, A/B/C, n

8/1/4

60/10/72

0.371

Combined fracture operation, n(%)

5(38.5)

9(6.3)

0.002

Ulnar nerve anterior transposition, n(%)

3(23.1)

73(51.4)

0.080

Operation , time, minutes

93.8±32.2

125.9±50.9

0.131

Table 4. Comparison of surgical approaches and fixation methods between non-union patients and union patients.

Surgical factors

Non-union

(n=13)

Union

(n=142)

P value

Surgical approach, n

Posterior

9

109

0.170

Medial

0

4

Lateral

2

25

Dual incision

2

4

Olecranon osteotomy/triceps split/paracipital‡

8/1/0

79/13/17

0.626

Fixation method, n(%)

Orthogonal plate

3(23.1)

84(59.2)

0.005

parallel

1(7.7)

7(4.9)

single

4(30.8)

40(28.2)

TBW

5(38.5)

11(7.7)

TBW, tension band wiring. ‡ Among the posterior incisions.

How did you manage non-union in this cohort`? Please elaborate.

Response: Thank you for your comment. We added additional treatment for 13 patients with nonunion.

Page 7, line 182-186: Of the 13 patients with nonunion, revision ORIF with auto bone graft was performed in eight patients and fracture union was obtained. One patient underwent total elbow arthroplasty, 1 patient had functional nonunion, 1 patient refused additional surgery, and 2 patients were lost to follow up while following up at an outpatient clinic after diagnosis of nonunion.

Where there any infections or infected non.-unions? How did you diagnose and treat those?

Response: Thank you for your comment. Infection was observed in one case, and there was no infected nonunion. We diagnosed infection by elevation of inflammatory marker (white blood cell count, white blood cell differential for segmented neutrophils, erythrocyte sedimentation rate, C-reactive protein) and fluid aspiration analysis at the operation site. The patient was treated with surgical debridement and intravenous antibiotics, and bone union was obtained. Since, it was not an infected nonunion patient, we did not mention it in the manuscript.

Discussion: Please provide more details on the studies you cite and where your study comes in. I still do not know what the studies by Vanclair or Claessen are about.

Response: Thank you for your comment. We provided more details on the studies of Vanclair and Claessen, based on the reviewer’s comment.

Page 7-8, line 197-200: Vanclair et al. reported that after ORIF of humerus fractures, nonunions occurred more frequently in the distal regions than in the proximal and diaphyseal regions, and  the incidence of nonunion after distal humerus fracture has been reported up to 25%.

Page 8, line 213-217: Claessen et al. analyzed the risk factors for reoperation after ORIF of distal humerus fractures due to early loosening or breakage of implants or non-union and reported that greater comorbidities, obesity, DM, and smoking were associated with reoperation . However, independent risk factors for nonunion were not evaluated in this study.

Please restructure the discussion.

One paragraph non-union rates in different studies

one paragraph patient risk factors

one paragraph surgical risk factors

one paragraph limitations. As discussed above there are quite a few more.

Response: Thank you for your comment. We restructured the discussion part according to the reviewer’s opinion. In addition, we added the limitation mentioned above by the reviewer.

Page 8, line 245-249: Third, conservative treatment can be performed in selected patients with distal humerus fractures. Since, this study was conducted on patients who underwent ORIF for distal humerus fracture, we could not evaluate the risk factors for nonunion that could occur after conservative treatment

We truly appreciate the time and effort dedicated toward reviewing this manuscript. We the authors appreciate reviewer’s insightful comments. We thank all reviewers for taking out of their time and energy to help improve our manuscript.

Thank you for your consideration. I look forward to hearing from you.

Round 2

Reviewer 2 Report

improved through revision